# Advancing Regulation in Artificial Intelligence: An Auction-Based Approach

## Abstract

In an era of "moving fast and breaking things", regulators have moved slowly to pick up the safety, bias, and legal debris left in the wake of broken Artificial Intelligence (AI) deployment. While there is much-warranted discussion about how to address the safety, bias, and legal woes of state-of-the-art AI models, rigorous and realistic mathematical frameworks to regulate AI are lacking. Our paper addresses this challenge, proposing an auction-based regulatory mechanism that provably incentivizes devices (i) to deploy compliant models and (ii) to participate in the regulation process. We formulate AI regulation as an all-pay auction where enterprises submit models for approval. The regulator enforces compliance thresholds and further rewards models exhibiting higher compliance than their peers. We derive Nash Equilibria demonstrating that rational agents will submit models exceeding the prescribed compliance threshold. Empirical results show that our regulatory auction boosts compliance rates by 20% and participation rates by 15% compared to baseline regulatory mechanisms, outperforming simpler frameworks that merely impose minimum compliance standards.

## 1 Introduction

The recent large-scale deployment of artificial intelligence (AI) models, such as large language models (LLMs), has simultaneously boosted human productivity while sparking concern over safety (*e.g.,* hallucinations, bias, and privacy (Huang et al., 2025)). Many industry leaders, such as Google and OpenAI, remain embroiled in controversy surrounding bias and misinformation (Brewster, 2024; Robertson, 2024; White, 2024), safety (Jacob, 2024; Seetharaman, 2024; White, 2023), as well as legality and ethics (Bruell, 2023; Metz et al., 2024; Moreno, 2023) in their development and deployment of LLMs. Furthermore, irresponsible LLM deployment risks the spread of misinformation or propaganda by adversaries (Barman et al., 2024; Neumann et al., 2024; Sun et al., 2024). To date, a consistent and industry-wide solution to oversee responsible AI deployment remains elusive.

Naturally, one solution to mitigate these dangers is to increase governmental regulation over AI deployment. In the United States, there have been some strides, on federal (House, 2023) and state levels (Information, 2024), to regulate the safety and security of large-scale AI systems (including LLMs). While these recent executive orders and bills highlight the necessity to develop safety standards and enact safety and security protocols, few details are offered. This follows a consistent trend of well-deserved scrutiny towards the lack of AI regulation without providing an answer on *how to develop rigorous and realistic mathematical frameworks to achieve AI regulation*.

We believe that a rigorous and realistic mathematical framework for AI regulation consists of four key pieces: **(a)** the ability to model and to analyze participant decisions, **(b)** the existence of an "optimal" participant equilibrium, **(c)** limited mathematical assumptions, and **(d)** straightforward implementation of the framework by a regulator. This work takes a first step towards unlocking each of these four keys, designing a regulatory framework to not only enforce strict compliance, *e.g.,* safety or ethical compliance, of deployed AI models, but simultaneously to incentivize the production of more compliant AI models.

Specifically, we **(a)** formulate the AI regulatory process as an *all-pay auction*, where agents (enterprises) submit their models to a regulator. This novel auction-based regulatory mechanism leverages a reward-payment protocol that **(b)** emits Nash Equilibria at which agents *deploy models that are more compliant than a prescribed threshold*. Analysis of our auction-based approach **(c)** requires few

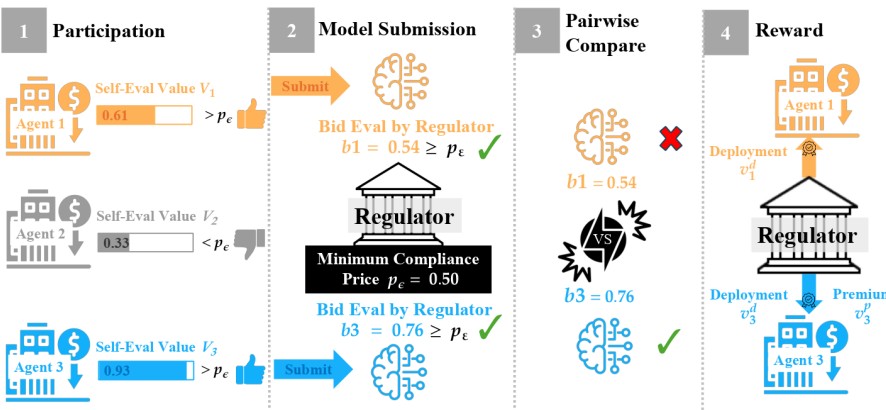

Figure 1: **Step-by-Step CIRCA Schematic.** (Step 0) The regulator sets a compliance threshold, $\epsilon$, having corresponding price, $p_\epsilon$, required to achieve $\epsilon$. (Step 1) Agents evaluate their total value, $V_i$, from model deployment value ($v_i^d$) and potential regulator compensation ($v_i^p$). Agents only participate if their total value exceeds $p_\epsilon$. (Step 2) Participating agents submit their models to the regulator, accompanied by their bid $b_i$, which reflects the amount spent to improve their model's compliance level. Models with bids below $p_\epsilon$ are automatically rejected. (Step 3) The submitted models are randomly paired, and the more compliant model (*i.e.,* the higher bid) in each pair wins. In this example, agent 3 wins since $b_3 > b_1$. (Step 4) Winning models receive both a premium and deployment value (*i.e.,* agent 3 wins premium $v_3^p$ and deployment $v_3^d$ values), while losing models receive only the deployment value (*i.e.,* agent 1 only wins deployment value $v_1^d$).

assumptions. While inclusion of assumptions is non-ideal, the usage of these assumptions allows us to advance AI regulation within a sparse, yet critical, area of research. We note, however, that the two assumptions used in this work are used within existing regulatory and AI settings (Goulder & Schein, 2013; Howe et al., 2024; Rajpurkar et al., 2016; Stavins, 2008; U.S. Food and Drug Administration, 2022; Williams et al., 2018; Zaremba et al., 2025) (Section 3). Finally, our approach is **(d)** simple and can easily be implemented by a regulator (Figure 1). Like existing regulatory frameworks (Coglianese & Kagan, 2007; Powell, 2014; Van Norman, 2016), we only require the regulator to: (i) prohibit deployment of models that fail to meet prescribed compliance thresholds, and (ii) incentivize compliant model production and deployment by providing additional rewards to agents that submit more compliant models than their peers.

We summarize our contributions as follows:

**(1) AI Regulation:** We propose a Compliance-Incentivized Regulatory-Centered Auction (CIRCA), offering a novel approach towards AI regulation.

**(2) Compliance-First:** We establish, through derived Nash Equilibria, that agents are incentivized to submit models surpassing the required compliance threshold.

**(3) Effectiveness:** Our empirical results show that CIRCA increases model compliance by over 20% and boosts participation rates by 15% compared to baseline regulatory mechanisms.

## 2 RELATED WORKS

**Regulation Frameworks for Artificial Intelligence.** A handful of work focuses on regulation frameworks for AI deployment (de Almeida et al., 2021; Jagadeesan et al., 2024; Rodríguez et al., 2022). First, de Almeida et al. (2021) details the need for AI regulation and surveys existing proposals. The proposals are ethical frameworks that express ethical decisions to make and dilemmas to address. However, these proposals lack a mathematical framework to incentivize provably compliant models. Rodríguez et al. (2022) utilize AI models to detect collusive auctions. This work is related to our paper but in reverse: Rodríguez et al. (2022) applies AI to regulate auctions and to ensure that they are not collusive. In contrast, our paper aims to use auctions to regulate AI deployment. Jagadeesan et al. (2024) focuses on reducing barriers to entry for smaller companies that are competing against

incumbent, larger companies. A multi-objective high-dimensional regression framework is proposed to impose "reputational damage" upon companies that deploy unsafe AI models. This work allows varying levels of safety constraints, where newer companies face less severe constraints in order to spur their entry into the market, which is unrealistic in many settings and only considers simple linear-regression models.

The closest related work to ours, Yaghini et al. (2024), proposes a regulation game for ensuring privacy and fairness that is formulated as a Stackelberg game. This game is a multi-agent optimization problem that is also multi-objective (for fairness and privacy). An equilibrium-search algorithm is presented to ensure that agents remain on the Pareto frontier of their objectives (although this frontier is estimated algorithmically). Notably, Yaghini et al. (2024) considers only one model builder (agent) and multiple regulators that provide updates to the agent's strategy. Here, a more realistic setup is considered, where there are multiple agents and a single regulator whose goal is to incentivize compliant model deployment. It falls out of the scope of a regulator's job to collaborate with agents to optimize their strategy. Furthermore, the mechanism proposed here is simple and efficient. No Pareto frontier estimation or multiple rounds of optimization are required.

**All-Pay Auctions.** Compared to the dearth of literature in regulatory frameworks for AI, all-pay auctions are well-researched (Amann & Leininger, 1996; Baye et al., 1996; Bhaskar, 2018; DiPalantino & Vojnovic, 2009; Gemp et al., 2022; Goeree & Turner, 2000; Siegel, 2009; Tardos, 2017). These works formulate specific all-pay auctions and determine their equilibria. Some works consider settings where agents have complete information about their rivals' bids (Baye et al., 1996) while others consider incomplete information, such as only knowing the distribution of agent valuations (Amann & Leininger, 1996; Bhaskar, 2018; Tardos, 2017). One major application of all-pay auctions are crowd-sourcing competitions. Many agents participate to win a reward, with those losing incurring a small cost for their time, effort, *etc.* DiPalantino & Vojnovic (2009) is one of the first works to model crowd-sourcing competitions as an all-pay auction. Further research, such as Gemp et al. (2022), have leveraged AI to design all-pay auctions for crowd-sourcing competitions. However, instead of crowd-sourcing, our paper formulates the AI regulatory process as an asymmetric and incomplete all-pay auction. Previous analysis in this setting (Amann & Leininger, 1996; Bhaskar, 2018; Tardos, 2017) is leveraged to derive Nash Equilibria.

# 3 REGULATORY COMPLIANCE OF ARTIFICIAL INTELLIGENCE

There exists a regulator $R$ with the compliance power to set and to enforce laws and regulations (*e.g.,* U.S. government regulation on lead exposure). The regulator wants to regulate AI model deployment, by ensuring that all models meet a compliance threshold $\epsilon \in (0, 1)$, *e.g.,* the National Institute for Occupational Safety and Health (NIOSH) regulates that N95 respirators filter out at least 95% of airborne particles. If a model does not reach the compliance threshold $\epsilon$, then it is deemed unsafe and the regulator bars deployment. On the other side, there are $n$ rational model-building agents. Agents seek to maximize their own benefit, or utility.

**Bidding & Evaluation**. By law, each agent $i$ must submit, or bid in auction terminology, its model $w_i \in \mathbb{R}^d$ for evaluation to the regulator before it can be approved for deployment. Let $S(w; x) : \mathbb{R}^d \to \mathbb{R}_+$ output a compliance level (the larger the better) for model $w$ given data $x$. In effect, each agent, given its own data $x_i$, bids a compliance level $s_i^A := S(w_i; x_i)$ to the regulator. Subsequently, the regulator, using its own data $x_R$, independently evaluates the agent's compliance level bid as $s_i^R := S(w_i; x_R)$. Agent and regulator evaluation data is assumed to be independent and identically distributed (IID) $x_i, x_R \sim \mathcal{D}$.

**Assumption 1.** *Agent and regulator evaluation data comes from the same distribution $x_i, x_R \sim \mathcal{D}$.*

This assumption is realistic, because agents and regulators typically rely on standardized data collection processes (U.S. Food and Drug Administration, 2022) or widely accepted datasets (Rajpurkar et al., 2016; Williams et al., 2018) for evaluation. This ensures a fair and unbiased assessment of compliance. For example, FDA guidelines detail that data collection should assess efficacy and safety across various subgroups, *e.g.,* demographics, while also not changing "baseline data collection determined by the clinical trial objectives" (U.S. Food and Drug Administration, 2022). In areas such as Natural Language Processing, common datasets, or benchmarks, are employed to consistently evaluate model comprehension (Rajpurkar et al., 2016; Williams et al., 2018), trustworthiness (Wang et al., 2023), and security (Munoz et al., 2024) across various models. Therefore, it is reasonable to

define agent $i$'s compliance level bid as $s_i := \mathbb{E}_{x \sim \mathcal{D}}[S(w_i; x)]$. The scenario where evaluation data may be non-IID is addressed within Appendix G.

In regulatory settings, like the NIOSH example, a scalar compliance metric is often used. If multiple compliance metrics must be monitored, $S$ can be defined to aggregate and weigh the various metrics. This too is realistic in AI. For example, LLM safety alignment literature uses a scalar-valued reward to ensure a model is aligned (Christiano et al., 2017; Kaufmann et al., 2023; Ouyang et al., 2022).

**Price of Compliance**. We assume that there exists a strictly increasing function $M : (0,1) \to (0,1)$ that determines the "price of compliance" (*i.e.,* maps compliance into cost). Simply put, higher-compliant models cost more to attain. Thus, we define the price of $\epsilon$-compliance as $p_\epsilon := M(\epsilon)$.

**Assumption 2.** $\epsilon > \epsilon' \implies M(\epsilon) > M(\epsilon')$. *A strictly increasing $M$ maps compliance to cost.*

One prominent existing example of this relationship is the cap-and-trade system that the Environmental Protection Agency exercises to combat pollution (Goulder & Schein, 2013; Stavins, 2008). Companies that pollute above a set emission threshold can reach compliance by purchasing allowances, or pollution deficits, from other compliant companies. Thus, pollution compliance is attained with greater cost. For an example within AI, models can achieve higher safety compliance through Machine Unlearning (Liu et al., 2024) or AI Alignment (Dai et al., 2024). However, such methods incur greater computational and data collection costs in exchange for improved compliance. Furthermore, it has been found empirically that larger models and longer inference attain higher levels of compliance in adversarial training, robustness transfer, and defense (Howe et al., 2024; Zaremba et al., 2025). However, larger models and longer inference increase training and inference costs. We validate the compliance-cost relationship empirically in Section 6.

**Agent Costs**. Realistically for agents, training a compliant model comes with added cost. Consequently, each agent $i$ must decide how much money to *bid*, or spend, $b_i$ to make its model compliant. By Assumption 2, the compliance level of an agent's model will be $s_i = M^{-1}(b_i)$.

**Agent Values**. *(1) Model deployment value $v_i^d$.* While it costs more for agents to produce compliant models, they gain value from having their models deployed. Intuitively, this can be viewed as the expected value $v_i^d$ of agent $i$'s model. The valuation for model deployment varies across agents (*e.g.,* Google may value having its model deployed more than Apple). *(2) Premium reward value $v_i^p$.* Beyond value for model deployment, the regulator can also offer additional, or premium, compensation valued as $v_i^p$ by agents (*e.g.,* tax credits for electric vehicle producers or Fast Track and Priority Review of important drugs by the U.S. Food & Drug Administration). The regulator provides additional compensation to agents whose models exceed the prescribed compliance threshold. However, the value of this compensation varies across agents due to differing internal valuations. It is unrealistic for the regulator to compensate all agents meeting the compliance threshold due to budget constraints. Therefore, additional rewards are limited to a top-performing half of agents surpassing the threshold. This ensures targeted compensation for agents enhancing compliance while maintaining feasibility for the regulator.

**Value Distribution**. The total value for each agent $i$ is defined as $V_i := v_i^d + v_i^p$, which represents the sum of the deployment value and premium compensation. Although these values may vary widely in practice, $\{V_i\}_{i=1}^n$ is normalized for all $n$ agents to be between 0 and 1 for analytical tractability, allowing a standardized range. Consequently, the price to achieve the compliance threshold $\epsilon$ is also normalized to fall within the $(0,1)$ interval, *i.e.,* $p_\epsilon \in (0,1)$. The scaling factor $\lambda_i \sim \mathcal{D}_\lambda(0, 1/2)$ dictates the proportion of total value allocated to deployment versus compensation. Therefore, (i) the deployment value is $v_i^d := (1 - \lambda_i)V_i$, and (ii) the premium compensation value is $v_i^p := \lambda_i V_i$. Both $V_i$ and $\lambda_i$ are private to each agent, though the distributions $\mathcal{D}_V$ and $\mathcal{D}_\lambda$ are known by participants. The maximum allowable factor is set at $\lambda_i = 1/2$, reflecting the realistic constraint that compensation should not exceed deployment value. Although Section 5 primarily considers $\lambda_i \leq 1/2$, theoretical extensions can be made for scenarios where $\lambda_i > 1/2$.

**All-Pay Auction Formulation**. Overall, agents face a trade-off: producing higher-compliant models garners value, via the regulator, but incurs greater costs. Furthermore, in order to attain the rewards detailed above, agents must submit a model with a compliance level at least as large as $\epsilon$. This problem is formulated as an *asymmetric all-pay auction* with *incomplete information* (Amann & Leininger, 1996; Bhaskar, 2018; Tardos, 2017). An all-pay auction is used since agents incur an unrecoverable cost, training costs, when submitting their model to regulators. The auction is

formulated as asymmetric with incomplete information since valuations $V_i$ are private and differ for each agent.

**Agent Objective**. The objective, for each model-building agent $i$, is to maximize its own utility $u_i$. Namely, each agent seeks to determine an optimal compliance level to bid to the regulator $b_i$. However, given the all-pay auction formulation, agents may need to take into account all other agents' bids $\boldsymbol{b}_{-i}$ in order to determine their optimal bid $b_i^*$,

$$b_i^* := \arg\max_{b_i} u_i(b_i; \boldsymbol{b}_{-i}). \tag{1}$$

A major portion of this paper is constructing an auction-based mechanism, thereby designing the utility of each agent, such that each participating agent maximizes its utility when each bids more than "the price to obtain the minimum compliance threshold", *i.e.*, $b_i^* > p_\epsilon$. To begin, a simple mechanism is provided, already utilized by regulators, that does not accomplish this goal, before detailing the auction-based mechanism CIRCA that provably ensures that $b_i^* > p_\epsilon$ for all agents.

## 4 RESERVE THRESHOLDING: BASE REGULATION

The simplest method to ensure model compliance is for the regulators to set a reserve price, or minimum compliance level. This mechanism is coined the *multi-winner reserve thresholding auction*, where the regulator awards a deployment reward, $v_i^d$, to each agent whose model meets or exceeds the compliance threshold $\epsilon$. Within this auction, each agent $i$'s utility is mathematically formulated as,

$$u_i(b_i; \boldsymbol{b}_{-i}) = \begin{cases} -b_i \text{ if } b_i < p_\epsilon, \\ v_i^d - b_i \text{ if } b_i \geq p_\epsilon. \end{cases} \tag{2}$$

However, the formulation above is ineffective at incentivizing greater than $\epsilon$-level compliance.

**Theorem 1** (Reserve Thresholding Nash Equilibrium). *Under Assumption 2, agents participating in Reserve Thresholding Equation 2 have an optimal bid and utility of,*

$$b_i^* = p_\epsilon, \quad u_i(b_i^*; \boldsymbol{b}_{-i}) = v_i^d - p_\epsilon, \tag{3}$$

*and submit models with the following compliance level,*

$$s_i^* = \begin{cases} \epsilon & \text{if } u_i(b_i^*; \boldsymbol{b}_{-i}) > 0, \\ 0 \text{ (no submission)} & \text{else.} \end{cases} \tag{4}$$

When a regulator implements reserve thresholding, as formally detailed in Theorem 1, agents exert minimal effort, submitting models that just meet the required compliance threshold $\epsilon$. While this approach ensures that all deployed models satisfy minimum compliance, it fails to encourage agents to build models with compliance levels exceeding $\epsilon$. Additionally, agents whose deployment rewards are less than the cost of achieving compliance, *i.e.*, $v_i^d < p_\epsilon$, lack incentive to participate in the regulatory process. That lack of incentive leads to reduced participation rates (Remark 1).

**Remark 1** (Lack of Incentive). *Each agent is only incentivized to submit a model with compliance $s_i^* = \epsilon$. Our goal is to construct a mechanism that incentivizes agents to build models that possess compliance levels exceeding the minimum threshold: $s_i^* > \epsilon$.*

## 5 COMPLIANCE-INCENTIVIZED REGULATION: AUCTION-BASED APPROACH

To alleviate the lack of incentives within simple regulatory auctions, such as the one in Section 4, we propose a regulatory all-pay auction that mandates an equilibrium where agents *submit models with compliance levels exceeding $\epsilon$*.

**Algorithm Description.** The core component of the auction is that agent compliance levels are randomly compared against one another, with the regulator rewarding those having the superior compliant model with premium compensation. Performing the randomization process multiple times reduces the likelihood of unfair outcomes. Only agents with models that achieve a compliance level of

---

**Algorithm 1** Compliance-Incentivized Regulatory-Centered Auction (CIRCA)

---

1: Each agent $i$ receives their total value $V_i$ and partition ratio $\lambda_i$ from "nature"
2: Agents determine their optimal bids $b_i^*$ and corresponding utility $u_i(b_i^*)$ ▷ via Corollaries 1 or 2
3: Agents decide to participate, the set of participating agents is $P = \{j \in [n] \mid u_j(b_j^*; \mathbf{b}_{-i}) > 0\}$
4: **for** participating agents $j \in P$ **do**
5:     Spend $b_j^*$ to build a model, with compliance $s_j = M^{-1}(b_j^*)$, and submit it to the regulator
6: **end for**
7: Regulator verifies compliance levels, clearing models for deployment when $s_j \geq \epsilon \; \forall j \in P$
8: Regulator pairs up models, awarding compensation to agents with the more compliant model

---

$\epsilon$ or higher are eligible to participate in the comparison process; models that do not meet this threshold are automatically rejected. The detailed algorithmic block of CIRCA is depicted in Algorithm 1.

**Agent Utility.** The utility for each agent $i$ is therefore defined as in Equation 5,

$$u_i(b_i; \mathbf{b}_{-i}) = \left(v_i^d + v_i^p \cdot 1_{(b_i > b_j)}\right) \cdot 1_{(b_i \geq p_\epsilon)} - b_i. \tag{5}$$

Per regulation guidelines, the compliance criteria of an accepted model must at least be $\epsilon$. Equation 5 dictates that values are only realized by each agent if its model produces a bid larger than the required cost to reach $\epsilon$-compliance, $1_{(b_i \geq p_\epsilon)}$. Furthermore, agents only realize additional compensation value $v_i^p$ from the regulator if their compliance level outperforms a randomly selected agent $j$, $1_{(b_i > b_j)}$. Any agent that bids $b_i = 1$ will automatically win and realize both $v_i^p$ and $v_j^w$. It is important to note that the cost that every agent incurs when building its model is sunk: if the model is not cleared for deployment, the cost $-b_i$ is still incurred. The agent utility is rewritten in a piece-wise manner below,

$$u_i(b_i; \mathbf{b}_{-i}) = \begin{cases} -b_i & \text{if } b_i < p_\epsilon, \\ v_i^d - b_i & \text{if } b_i \geq p_\epsilon \text{ and } b_i < b_j \text{ random bid } b_j, \\ v_i^d + v_i^p - b_i & \text{if } b_i \geq p_\epsilon \text{ and } b_i > b_j. \end{cases} \tag{6}$$

By introducing additional compensation, $v_i^p$, and, crucially, conditioning it on whether an agent's model is more compliant than that of another random agent, it becomes rational for agents to bid more than the price to obtain the minimum compliance threshold (unlike Theorem 1).

**Incentivizing Agents to Build Compliant Models.** We establish a guarantee that agents participating in CIRCA *maximize their utility with an optimal bid $b_i^*$ that is larger than "the price required to attain $\epsilon$ compliance" (i.e., $b_i^* > p_\epsilon$)* in Theorem 2 below. Furthermore, agents bid in proportion to the value for additional compensation $v_i^p$ that the regulator offers for extra-compliant models.

**Theorem 2.** *Agents participating in CIRCA Equation 6 follow an optimal bidding strategy $\hat{b}_i^*$ of,*

$$\hat{b}_i^* := p_\epsilon + v_i^p F_v(v_i^p) - \int_0^{v_i^p} F_v(z) dz > p_\epsilon, \tag{7}$$

*where $F_v(\cdot)$ denotes the cumulative distribution function of the random premium reward variable corresponding to the premium reward $v_i^p = V_i \lambda_i$.*

Theorem 2 applies to any distribution for $V_i$ and $\lambda_i$ on $[0, 1]$ and $[0, 1/2]$, i.e., $V_i \sim \mathcal{D}_V(0, 1)$ and $\lambda_i \sim \mathcal{D}_\lambda(0, 1/2)$, respectively. Determining specific optimal bids, utility, and model compliance levels requires given distributions for $V_i$ and $\lambda_i$. Analysis of all-pay auctions (Amann & Leininger, 1996; Bhaskar, 2018; Tardos, 2017), as well as many other types of auctions, often assume a Uniform distribution for valuations. Therefore, our first analysis of CIRCA, below in Corollary 1, presumes Uniform distributions for $V_i$ and $\lambda_i$.

**Remark 2** (Improved Model Compliance). *Participating agents will submit models that are more compliant than the regulator's compliance threshold, $s_i^* = M^{-1}(b_i^*) > \epsilon$.*

**(Special Case 1) Uniform $V_i$ and $\lambda_i$: Optimal Agent Strategy.** Corollary 1 determines that a participating agent's optimal strategy to maximize its utility is to submit a model with compliance levels larger than $\epsilon$ when their values $V_i$ and $\lambda_i$ come from a Uniform distribution.

**Corollary 1** (Uniform Nash Bidding Equilibrium). *Under Assumption 2, for agents having total value $V_i$ and scaling factor $\lambda_i$ both stemming from a Uniform distribution, with $v_i^d = (1 - \lambda_i)V_i$, and*

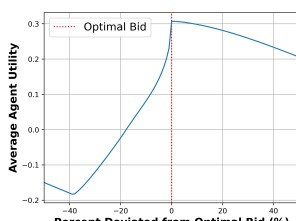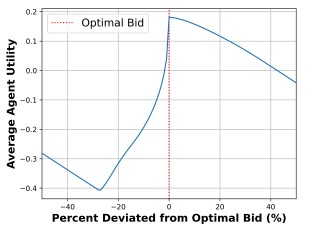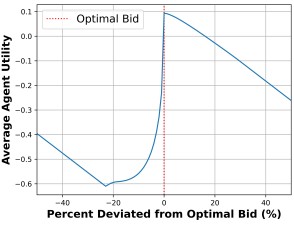

Figure 2: **Validation of Uniform Nash Bidding Equilibrium.** Agent utility is maximized when agents follow the theoretically optimal bidding function shown in Equation equation 8. Across varying compliance prices, $p_\epsilon = 0.25$ (left), $0.5$ (middle), $0.75$ (right), agents attain less utility when they deviate from the optimal bid (red line) derived in Corollary 1.

$v_i^p = \lambda_i V_i$, *their optimal bid and utility participating in* CIRCA *Equation 6 are* $b_i^* := \min\{\hat{b}_i^*, 1\}$,

$$\hat{b}_i^* = \begin{cases} p_\epsilon + \frac{(v_i^p)^2 \ln(p_\epsilon)}{p_\epsilon - 1} & \text{if } 0 \le v_i^p \le \frac{p_\epsilon}{2}, \\ p_\epsilon + \frac{8(v_i^p)^2(\ln(2v_i^p) - 1/2) + p_\epsilon^2}{8(p_\epsilon - 1)} & \text{if } \frac{p_\epsilon}{2} \le v_i^p \le \frac{1}{2}, \end{cases} \tag{8}$$

$$u_i(b_i^*; \boldsymbol{b}_{-i}) = \begin{cases} \frac{2(v_i^p)^2 \ln(p_\epsilon)}{p_\epsilon - 1} + v_i^d - b_i^* & \text{if } 0 \le v_i^p \le \frac{p_\epsilon}{2}, \\ \frac{2(v_i^p)^2(\ln(2p_\epsilon) - 1) + p_\epsilon}{p_\epsilon - 1} + v_i^d - b_i^* & \text{if } \frac{p_\epsilon}{2} \le v_i^p \le \frac{1}{2}. \end{cases} \tag{9}$$

*Participating agents submit models with compliance,*

$$s_i^* := \begin{cases} M^{-1}(b_i^*) > \epsilon & \text{if } u_i(b_i^*; \boldsymbol{b}_{-i}) > 0, \\ 0 \text{ (no submission)} & \text{else.} \end{cases} \tag{10}$$

**(Special Case 2): Beta $V_i$ and Uniform $\lambda_i$: Optimal Agent Strategy.** In many instances, a realistic distribution for $V_i$ is a Beta distribution with $\alpha, \beta = 2$. This distribution is Gaussian-like, with the bulk of the probability density placed in the middle. As such, it is realistic when agent values do not congregate amongst one another and outliers (near 0 or 1) are rare. The performance of CIRCA in this setting is analyzed in Corollary 2. Corollary 2 states that, under a Beta(2,2) distribution for $V_i$, agent $i$ maximizes its utility with an optimal bid $b_i^*$ larger than the price of $\epsilon$ compliance, $b_i^* > p_\epsilon$, resulting in a model above the $\epsilon$-compliance threshold. Furthermore, Corollaries 1 and 2 surpass the baseline optimal bid $b_i^* = p_\epsilon$ for Reserve Thresholding (Theorem 1).

**Corollary 2** (Beta Nash Bidding Equilibrium). *Under Assumption 2, let agents have total value $V_i$ and scaling factor $\lambda_i$ stem from Beta $(\alpha, \beta = 2)$ and Uniform distributions respectively, with $v_i^d = (1 - \lambda_i)V_i$ and $v_i^p = \lambda_i V_i$. Denote the CDF of the Beta distribution on $[0, 1]$ as $F_\beta(x) = 3x^2 - 2x^3$. The optimal bid and utility for agents participating in* CIRCA *Equation 6 are* $b_i^* := \min\{\hat{b}_i^*, 1\}$,

$$\hat{b}_i^* = \begin{cases} p_\epsilon + \frac{3(v_i^p)^2(p_\epsilon^2 - 2p_\epsilon + 1)}{1 - F_\beta(p_\epsilon)} & 0 \le v_i^p \le \frac{p_\epsilon}{2}, \\ p_\epsilon + \frac{8(v_i^p)^2\left(6(v_i^p)^2 - 8v_i^p + 3\right) + p_\epsilon^3(3p_\epsilon - 4)}{8(1 - F_\beta(p_\epsilon))} & \frac{p_\epsilon}{2} \le v_i^p \le 1/2, \end{cases} \tag{11}$$

$$u(b_i^*; \boldsymbol{b}_{-i}) = \begin{cases} v_i^d + \frac{6(v_i^p)^2(p_\epsilon^2 - 2p_\epsilon + 1)}{1 - F_\beta(p_\epsilon)} - b_i^* & 0 \le v_i^p \le \frac{p_\epsilon}{2}, \\ v_i^d + \frac{v_i^p\left(8(v_i^p)^3 - 12(v_i^p)^2 + 6v_i^p + p_\epsilon^2(2p_\epsilon - 3)\right)}{1 - F_\beta(p_\epsilon)} - b_i^* & \frac{p_\epsilon}{2} \le v_i^p \le 1/2. \end{cases} \tag{12}$$

*Participating agents submit models with compliance,*

$$s_i^* = \begin{cases} M^{-1}(b_i^*) > \epsilon & \text{if } u_i(b_i^*; \boldsymbol{b}_{-i}) > 0, \\ 0 \text{ (no submission)} & \text{else.} \end{cases} \tag{13}$$

**Remark 3** (Improved Utility & Participation). *Through introduction of premium compensation, agent utility is improved, in Equations 9 and 12, versus Reserve Thresholding in Equation 3. As a result, more agents break the zero-utility barrier of entry for participation, boosting both overall agent utility and participation rate.*

The proofs of Theorems 1 and 2 as well as Corollaries 1 and 2 are found within Appendix D. Since the premium compensation value $v_i^p$ is a product of two random variables, the PDF and CDF of $v_i^p$ becomes a piece-wise function (as shown within Appendix D). As a result, the optimal bidding and subsequent utility also becomes piece-wise in both Corollaries 1 and 2. Empirically, the correctness of the computed PDF and CDFs are verified within Appendix E.

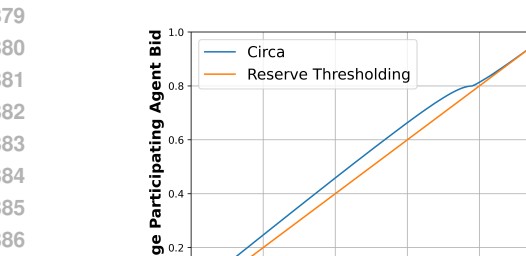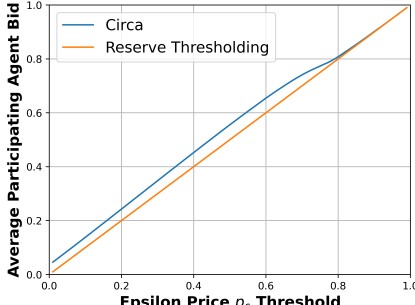

Figure 3: **Improved Compliance with Uniform & Beta Values**. When total value stems from a (left) Uniform $V_i \sim U(0, 1)$ or (right) Beta distribution $V_i \sim \text{Beta}(\alpha = \beta = 2)$, agents bid more compliant models in CIRCA than Reserve Thresholding.

## 6 EXPERIMENTS

Section 5 demonstrates that CIRCA creates incentives for any agents to submit compliant-exceeding models and to participate at rates higher than the baseline Reserve Thresholding mechanism in Section 4. Below, we validate these theoretical results empirically.

**Experimental Setup**. A regulatory setting with $n = 100,000$ agents is simulated below. Each agent $i$ receives a random total value $V_i$ from either a Uniform (Corollary 1) or Beta(2,2) (Corollary 2) distribution. Each agent also receives a scaling factor $\lambda_i$ that splits the total value into deployment $v_i^d = (1 - \lambda_i)V_i$ and premium compensation $v_i^p = \lambda_i V_i$ values. Once private values are provided, agents calculate their bid according to the optimal strategies in Theorems 1, 2 and Corollaries 1, 2.

**Lack of Baseline Regulatory Mechanisms**. To the best of knowledge, there are no other comparable compliance mechanisms to regulate AI. As a result, the Reserve Threshold mechanism that is proposed in Section 4 is used as a baseline. While simple, the Reserve Threshold mechanism is a realistic baseline to compare against. For example, existing regulatory bodies, like the Environmental Protection Agency (EPA), follow similar steps before clearing products (*e.g.,* the EPA authorizes permits for discharging pollutants into water sources once water quality criteria are met).

**Verifiable Nash Bidding Equilibria**. The first experimental goal is to validate that the theoretical bidding functions found in Corollaries 1 and 2 constitute Nash Equilibria. That is, an agent receives worse utility if it deviates from this bidding strategy if other agents abide by it. To test this, the optimal bid for a single agent is compared versus $100,000$ others. The single agent's optimal bid is varied on a range up to $\pm 50\%$. Note that comparisons only occur if the other agent's bid is at least $p_\epsilon$, in order to accurately reflect how the auction mechanism in Algorithm 1 functions.

In Figures 2 and 8 (Appendix E), the average utility over all $100,000$ comparisons is plotted. *One can see that both the Uniform and Beta optimal-bidding functions maximize agent utility and thus constitute Nash Equilibria*. Utility decays much quicker when reducing the bid, since agents are **(i)** less likely to win the premium reward and **(ii)** at risk of losing the value from deployment if the bid does not reach $p_\epsilon$. At a certain point, utility increases linearly once the agent continuously fails to bid $p_\epsilon$. The linear improvement stems from the agent saving the cost of its bid, $-b_i$, shown in Equation 6.

**Improved Agent Participation and Bid Size**. For both Uniform and Beta(2,2) distributions, shown in Figures 3 and 4, the proposed mechanism (CIRCA) increases participation rates and average bids by upwards of 15% and 20% respectively. At the endpoints of possible price thresholds, $p_\epsilon = 0$ and 1, both mechanisms perform similarly. The reason is that at a low compliance threshold price $p_\epsilon \approx 0$, agents are highly likely to have a total value $V_i$ larger than a value close to zero. The inverse is true for $p_\epsilon \approx 1$, where it is unlikely that agents will have $V_i$ larger than a value close to 1. The proposed mechanism shines when compliance threshold prices are in the middle; the premium compensation offered by the regulator incentivizes agents to participate and bid more for the chance to win.

**Compliance-Cost Case Study**. Below, a case study is conducted to demonstrate that in realistic settings, compliance is mapped to cost in a monotonically increasing way (as de-

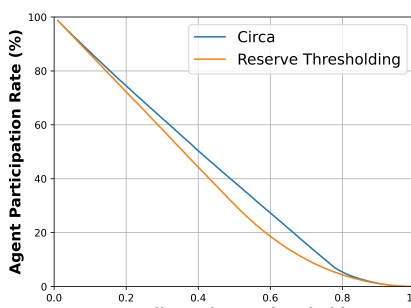 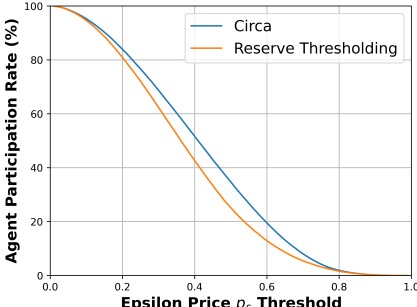

Figure 4: **Improved Participation with Uniform & Beta Values**. When total value stems from a (left) Uniform $V_i \sim U(0,1)$ or (right) Beta distribution $V_i \sim \text{Beta}(\alpha = \beta = 2)$, agents participate at a higher rate in CIRCA than Reserve Thresholding.

tailed in Assumption 2). While there are many compliance metrics to consider when gauging AI deployment, model fairness is analyzed, via equalized odds, for image classification in this study. Equalized odds measures if different groups have similar true positive rates and false positive rates (lower is better). Multiple VGG-16 models are trained on the Fairface dataset (Karkkainen & Joo, 2021) for fifty epochs (repeated ten times with different random seeds), and consider a gender classification task with race as the sensitive attribute. Models with the largest validation classification accuracy during training are selected for testing.

Many types of costs exist for training compliant models, such as extensive architecture and hyper-parameter search. In this study, the cost of an agent acquiring more minority class data is considered. Acquiring more minority class data leads to a larger and more balanced dataset. Various mixtures of training data are simulated, starting from a 95:5 skew and scaling up to fully balanced training data with respect to the sensitive attribute. In this study, equalized odds performance is gauged on well-balanced test data for the models trained on various mixtures of data. The results of this case study are shown in Figure 5 and Table 3 (Appendix E).

As expected, in Table 3, the equalized odds score decreases (more compliant model) when collecting more minority class data (increased cost). To adjust equalized odds to fit into the setting where $\epsilon \in (0,1)$,

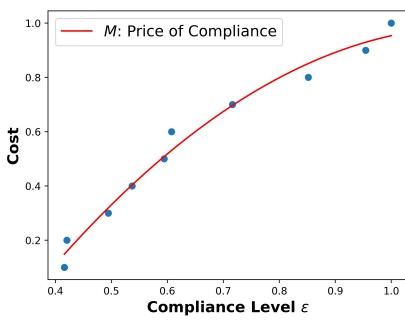

Figure 5: **Strictly Monotonic Compliance-Cost Relationship.** As the percentage of minority class data increases (greater cost), equalized odds metric improves (greater compliance) on Fairface.

the original equalized odds score are inverted and normalized. In Figure 5, one can see that compliance level is indeed monotonically increasing with respect to the cost.

## 7 CONCLUSION

As AI models grow, the risks associated with their misuse become significant, particularly given their opaque, black-box nature. Establishing robust algorithmic safeguards is crucial to protect users from unethical, unsafe, or illegally-deployed models. In this paper, we present a regulatory framework designed to ensure that only models deemed compliant by a regulator can be deployed for public use. Our key contribution is the development of an auction-based regulatory mechanism that simultaneously (i) enforces compliance standards and (ii) provably incentivizes agents to exceed minimum compliance thresholds. This approach encourages broader participation and the development of more compliant models compared to baseline regulatory methods. Empirical results confirm that our mechanism increases agent participation by 15% and raises agent spending on compliance by 20%, demonstrating its effectiveness to promote more compliant AI deployment.

## ETHICS & IMPACT STATEMENT

Unchecked AI deployment runs the risk of unsafe consequences that can harm users and stoke division within our society. It is imperative to outline and employ regulatory frameworks to mitigate these dangers and ensure user safety. However, regulation in AI is heavily under-researched. The goal of this paper is to take a step towards designing realistic and effective regulation to ensure AI model compliance. We hope that the impact of our paper will spur future research into regulatory AI, and soon provide a robust solution for governments to implement.

## REPRODUCIBILITY STATEMENT

As this paper is mainly theoretical in nature, our reproducibility statement pertains to the assumptions and proofs used to derive our Nash Equilibria. In Section 3, we introduce both of our assumptions and detail why they are justifiable. In Appendix D, proofs of Theorems 1 and 2 as well as Corollaries 1 and 2 are well-detailed. Steps of all proofs are carefully documented to ensure that a reader can reproduce our theoretical results on their own. Finally, we have provided the code for our experimental results for viewing and reproduction. This code will become open-sourced after publication of the paper.

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

## A   Summary and Comparison of Contributions

Table 1: Comparison of AI Regulation Frameworks

| Feature | CIRCA (This Paper) | Jagadeesan et al. (2024) | Yaghini et al. (2024) | All-Pay Auctions (General) |
|---|---|---|---|---|
| Overview | Formulates AI regulation as an auction to derive Nash Equilibria. | Penalizes larger companies more for unsafe AI models. | Introduces a multi-agent, multi-objective regulatory game. | Diverse formulations of all-pay auctions. |
| Regulatory Scheme? | ✓ | ✗ | ✓ | ✗ |
| Compliance-Aware Mechanism? | ✓ | ✗ | ✓ | ✗ |
| Theoretical Guarantees? | ✓ | ✗ | ✗ | ✓ |
| Incentivizing Over-Compliance? | ✓ | ✗ | ✗ | ✗ |
| Multiple Model Builders? | ✓ | ✓ | ✗ | ✓ |
| Single Round (Simple)? | ✓ | ✓ | ✗ | ✓ |

CIRCA introduces novel theoretical analysis of compliance-aware, all-pay auctions. Here are some of the highlights:

1. Our main technical contribution is the introduction and equilibrium analysis of a compliance-aware, multi-tiered all-pay auction, which has not been previously studied. While traditional all-pay auctions have been explored in economic theory, prior work (*e.g.,* Jagadeesan et al. (2024), Yaghini et al. (2024)) either does not target compliance, lacks theoretical guarantees, or does not use comparison-based mechanisms. Table 1 outlines these distinctions in detail.

2. Specifically, we introduce two key theoretical contributions:

   - Theorem 1: Equilibrium analysis of a reserve-threshold-modified all-pay auction.
   - Theorem 2: Equilibrium analysis under a novel pairwise comparison mechanism (CIRCA) that rewards the more compliant of two randomly selected agents.

3. We also provide generalizability proofs under realistic value distributions (Corollaries 1 and 2). These results go beyond derivations and reflect new applications of game-theoretic reasoning to regulatory.

## B NOTATION TABLE

Table 2: Notating and Defining all Variables Listed Within CIRCA.

| Definition | Notation |
|---|---|
| **Regulator** | $R$ |
| **Number of Agents** | $n$ |
| **Compliance Threshold** | $\epsilon$ |
| **Compliance-to-Cost Function** | $M$ |
| **Price of Attaining Compliance** | $p_\epsilon$ |
| **Agent $i$ Bid** | $b_i$ |
| **Agent $i$'s Optimal Bid** | $b_i^*$ |
| **All Other Agents Bids** | $\boldsymbol{b}_{-i}$ |
| **Agent $i$ Utility** | $u_i$ |
| **Agent $i$ Model Compliance** | $s_i$ |
| **Total Value for Agent $i$** | $V_i$ |
| **Total Value Distribution** | $\mathcal{D}_V$ |
| **Agent $i$ Scaling Factor** | $\lambda_i$ |
| **Scaling Factor Distribution** | $\mathcal{D}_\lambda$ |
| **Deployment Value for Agent $i$** | $v_i^d$ |
| **Premium Compensation Value for Agent $i$** | $v_i^p$ |
| **Probability Density Function for Premium Compensation** | $f_v$ |
| **Cumulative Distribution Function for Premium Compensation** | $F_v$ |

## C BINARY AND DISCRETE COMPLIANCE IN CIRCA

Our framework still works within binary or discrete settings. This is important when dealing with properties or metrics that are not continuous, like how the EU AI Act evaluates AI risk into minimal, limited, high, and unacceptable tiers (Act, 2024). The rationale behind why CIRCA works for binary or discrete settings is that models can still be ranked or compared against each other depending on how well they satisfy the given metric or property.

For example, models can be separated into Pass/Fail categories, where the Pass category can be further split into High/Medium/Low sub-categories. All models achieving at least Low Pass compliance are cleared for deployment. While a model either complies or does not, the models can still be gauged on how well they comply (e.g., High/Medium/Low). Since a ranking of models can still be generated, premium rewards can be provided to higher-passing models.

In situations where the regulatory policy is black and white, for example "your model must be trained with differential privacy", CIRCA still holds as an ordering or ranking between models can still be ascertained. In the example of differential privacy, *any* model that is trained with differential privacy would be cleared for deployment. However, it is also true that differential privacy can be gauged by a level of privacy $\epsilon_{DP}$ (not to be confused with our compliance threshold $\epsilon$). Models with smaller values of $\epsilon_{DP}$ will be provided additional premium rewards since they are more compliant (*i.e.,* more private). Thus, CIRCA would still incentivize agents to become more private even when there is a binary compliance metric.

# D THEORETICAL PROOFS

Below, we provide the full proofs of our Theorems and Corollaries presented within our work.

## D.1 PROOF OF THEOREM 1

**Theorem 1** (Restated). *Under Assumption 2, agents participating in Reserve Thresholding Equation 2 have an optimal bid and utility of,*

$$b_i^* = p_\epsilon, \quad u_i(b_i^*; \boldsymbol{b}_{-i}) = v_i^d - p_\epsilon,$$

*and submit models with the following compliance level,*

$$s_i^* = \begin{cases} \epsilon & \text{if } u_i(b_i^*; \boldsymbol{b}_{-i}) > 0, \\ 0 \text{ (no submission)} & \text{else.} \end{cases}$$

*Proof.* From agent $i$'s utility within Reserve Thresholding, Equation 2, it is clear that $u_i(0) = 0$. We proceed to break the proof up into cases where agents have (1) a deployment value equal to or less than the price of compliance $p_\epsilon$ and (2) a deployment value larger than $p_\epsilon$.

**Case 1:** $v_i^d \leq p_\epsilon$. From Equation 2, if $v_i^d \leq p_\epsilon$ then an agent will never attain positive utility,

$$\max_{b_i \in (0,1]} v_i^d \cdot 1_{b_i \geq p_\epsilon} - b_i \leq \max_{b_i \in (0,1]} p_\epsilon \cdot 1_{b_i \geq p_\epsilon} - b_i = \max_{b_i \in [p_\epsilon, 1]} p_\epsilon - b_i = p_\epsilon - p_\epsilon = 0. \quad (14)$$

$$\arg\max_{b_i \in (0,1]} u_i(b_i) = p_\epsilon. \quad (15)$$

For an agent with deployment value at most equal to $p_\epsilon$, the upper bound on attainable utility when it participates, *i.e.,* $b_i \in (0,1]$, is zero (Equation 14). This maximum utility is attained when bidding $b_i = p_\epsilon$ (Equation 15). Thus, agents have nothing to gain by participating, as they already start at zero utility $u_i(0) = 0$. As a result, agents will not submit a model, $s_i^* = M(0) = 0$.

**Case 2:** $v_i^d > p_\epsilon$. Similar steps to Case 1 above,

$$\max_{b_i \in (0,1]} v_i^d \cdot 1_{b_i \geq p_\epsilon} - b_i > \max_{b_i \in (0,1]} p_\epsilon \cdot 1_{b_i \geq p_\epsilon} - b_i = \max_{b_i \in [p_\epsilon, 1]} p_\epsilon - b_i = p_\epsilon - p_\epsilon = 0. \quad (16)$$

$$b_i^* = \arg\max_{b_i \in (0,1]} u_i(b_i) = p_\epsilon \longrightarrow u_i(b_i^*) = v_i^d - p_\epsilon > 0. \quad (17)$$

An agent with deployment value larger than $p_\epsilon$ will have a maximal utility that is non-negative when it participates (Equation 16). Maximal utility is attained when bidding $b_i^* = p_\epsilon$ (Equation 17). Furthermore, at this optimal bid, the corresponding compliance level is $s_i^* = M^{-1}(p_\epsilon) = \epsilon$.

$\square$

## D.2 PROOF OF THEOREM 2

**Theorem 2** (Restated). *Agents participating in* CIRCA *Equation 6 will follow an optimal bidding strategy $\hat{b}_i^*$ of,*

$$\hat{b}_i^* := p_\epsilon + v_i^p F_v(v_i^p) - \int_0^{v_i^p} F_v(z)dz > p_\epsilon,$$

*where $F_v(\cdot)$ denotes the cumulative distribution function of the random premium reward variable corresponding to the premium reward $v_i^p = V_i \lambda_i$.*

*Proof.* Before beginning our proof, we note that each agent $i$ cannot alter its own valuation $v_i^p$ for winning the all-pay auction. Each valuation is private (unknown by other agents) and predetermined: total reward $V_i$ and partition factor $\lambda_i$ are randomly selected from a given distribution $\mathcal{D}$ on $[0,1]$ and $[0, 1/2]$ respectively by "nature". We define the cumulative distribution function for the auction reward $v_i^p = V_i \lambda_i$ as $F_v(\cdot)$ and the probability distribution function as $f_v(\cdot)$.

From Equation 6, we find that an agent $i$ that does not participate (*i.e.,* $b_i = 0$) receives no utility,

$$u_i(0) = 0. \quad (18)$$

An agent receives negative utility if its bid does not reach the price of compliance $p_\epsilon$,

$$\max_{b_i \in (0, p_\epsilon)} u_i(b_i) < 0. \tag{19}$$

Consequently, rational agents will either opt not to participate (notated as the set of agents $N$) or participate (notated as the set of agents $P$) and bid at least $p_\epsilon$. We define these groups as,

$$N = \{i \in [n] \mid \max_{b_i \in [0,1]} u_i(b_i) \leq 0\}, \tag{20}$$

$$P = \{i \in [n] \mid \max_{b_i \in [0,1]} u_i(b_i) > 0\}. \tag{21}$$

From here, we only focus on agents $i \in P$ which participate (*i.e.,* have utility to be gained by participating). As a result from Equations 18 and 19, Equation 21 transforms into,

$$P = \{i \in [n] \mid \max_{b_i \in [p_\epsilon, 1]} u_i(b_i) > 0\}. \tag{22}$$

The result of Equation 22 is that participating agents bid at least $p_\epsilon$. This is important, as every participating agent knows that all rival agents $j$ they will possibly be compared against have $b_j \in [p_\epsilon, 1]$. Agents can dictate how much they bid, and we design our auction to ensure that agents bid in proportion to their valuation.

Following previous literature (Amann & Leininger, 1996; Bhaskar, 2018; Tardos, 2017), we desire a *monotone increasing* bidding function $b(\cdot) : [0, 1/2] \to [p_\epsilon, 1]$ that each agent follows. We will prove that each agent $i$'s best strategy is to bid its own valuation $b(v_i^p)$ irrespective of other agent bids (Nash Equilibrium). Using a bidding function transforms agent utility,

$$u_i(b_i) = \left(v_i^d + v_i^p \cdot 1_{(\text{if } i \text{ wins auction})}\right) \cdot \underbrace{1_{(\text{if } b_i \geq p_\epsilon)}}_{\text{satisfied for agents } i \in P} - b_i,$$

$$= \mathbb{P}\left(b(b_i) > b(b_j)\right)v_i^p - b(b_i) + v_i^d, \quad b_j \sim \text{randomly sampled agent bid.} \tag{23}$$

Since $b(x)$ is monotone increasing up to 1, agents bidding $b = 1$ automatically win, the utility function above can be simplified as,

$$u_i(b_i) = v_i^p \mathbb{P}\left(b_i > b_j\right) - b(b_i) + v_i^d, \quad b_j \sim \text{randomly sampled agent bid,}$$

$$= v_i^p F_v(b_i) - b(b_i) + v_i^d. \tag{24}$$

Taking the derivative and setting it equal to zero yields,

$$\frac{d}{db_i} u_i(b_i) = v_i^p f_v(b_i) - b'(b_i) = 0. \tag{25}$$

As agents bid in proportion to their valuation, we solve the first-order equilibrium conditions at $b_i = v_i^p$,

$$b'(v_i^p) = v_i^p f_v(v_i^p). \tag{26}$$

Integrating by parts, and knowing $\epsilon$ is the minimum bid ($b(0) = p_\epsilon$), reveals our optimal bidding function,

$$b(v_i^p) - b(0) = \int_0^x v_i^p f_v(v_i^p) dv_i^p,$$

$$b(v_i^p) - p_\epsilon = v_i^p F_v(v_i^p) - \int_0^{v_i^p} F_v(z) dz,$$

$$\hat{b}_i^* = b(v_i^p) := p_\epsilon + v_i^p F_v(v_i^p) - \int_0^{v_i^p} F_v(z) dz. \tag{27}$$

$\square$

### D.3 PROOF OF COROLLARY 1

**Corollary 1** (Restated). *Under Assumption 2, for agents having total value $V_i$ and scaling factor $\lambda_i$ both stemming from a Uniform distribution, with $v_i^d = (1 - \lambda_i)V_i$, and $v_i^p = \lambda_i V_i$, their optimal bid and utility participating in* CIRCA *Equations 6 are $b_i^* := \min\{\hat{b}_i^*, 1\}$,*

$$\hat{b}_i^* = \begin{cases} p_\epsilon + \frac{(v_i^p)^2 \ln(p_\epsilon)}{p_\epsilon - 1} & \text{if } 0 \le v_i^p \le \frac{p_\epsilon}{2}, \\ p_\epsilon + \frac{8(v_i^p)^2(\ln(2v_i^p) - 1/2) + p_\epsilon^2}{8(p_\epsilon - 1)} & \text{if } \frac{p_\epsilon}{2} \le v_i^p \le \frac{1}{2}, \end{cases}$$

$$u_i(b_i^*; \boldsymbol{b}_{-i}) = \begin{cases} \frac{2(v_i^p)^2 \ln(p_\epsilon)}{p_\epsilon - 1} + v_i^d - b_i^* & \text{if } 0 \le v_i^p \le \frac{p_\epsilon}{2}, \\ \frac{2(v_i^p)^2(\ln(2p_\epsilon) - 1) + p_\epsilon}{p_\epsilon - 1} + v_i^d - b_i^* & \text{if } \frac{p_\epsilon}{2} \le v_i^p \le \frac{1}{2}. \end{cases}$$

*Participating agents submit models with compliance,*

$$s_i^* := \begin{cases} M^{-1}(b_i^*) > \epsilon & \text{if } u_i(b_i^*; \boldsymbol{b}_{-i}) > 0, \\ 0 \text{ (no submission)} & \text{else}. \end{cases}$$

*Proof.* Let $v_i^p := V_i \lambda_i$, where $V_i \sim U[p_\epsilon, 1]$ and $\lambda_i \sim U[0, 1/2]$. The reason that $V_i$ is within the interval $[p_\epsilon, 1]$, is that all participating agents must have a value of at least $p_\epsilon$ or else they would not have rationale to bid. The smallest value of $V_i$ such that this is possible is $p_\epsilon$, so it is the lower bound on this interval. Our first goal is to find the PDF of $v_i^p$, $f_{v_i^p}(\cdot)$.

We begin solving for $f_{v_i^p}(\cdot)$ by using a change of variables. For the product of two random variables $v = x_1 \cdot x_2$, let $y_1 = x_1 \cdot x_2$ and $y_2 = x_2$. Thus, we find inversely that $x_2 = y_2$ and $x_1 = y_1/y_2$. Since $x_1$ and $x_2$ are independent and both uniform, we find that,

$$f_{y_1, y_2}(x_1, x_2) = (\frac{1}{1 - p_\epsilon})(\frac{1}{1/2 - 0}) = \frac{2}{1 - p_\epsilon}. \tag{28}$$

When using the change of variables this becomes,

$$f_{y_1, y_2}(y_1, y_2) = f_{y_1, y_2}(x_1, x_2)|J| = \frac{2}{(1 - p_\epsilon)y_2}, \quad |J| = \left| \begin{pmatrix} 1/y_2 & -y_1/y_2^2 \\ 0 & 1 \end{pmatrix} \right| = 1/y_2 \tag{29}$$

Marginalizing out $y_2$ (a non-negative value) yields,

$$f_{y_1}(y_1) = \int_0^\infty \frac{2}{(1 - p_\epsilon)y_2} dy_2. \tag{30}$$

The bounds of integration depend upon the value of $y_1$. The change of variable to the $(y_1, y_2)$ space, where $0 \le y_1, y_2 \le 1/2$, results in a new region of possible variable values. This region is a triangle bounded by the three vertices: $(0, 0)$, $(p_\epsilon/2, 1/2)$, and $(1/2, 1/2)$. Thus, the bounds of marginalization depend upon the value of $y_1$. For $0 \le y_1 \le p_\epsilon/2$ we have,

$$f_{y_1}(y_1) = \int_{y_1}^{y_1/p_\epsilon} \frac{2}{(1 - p_\epsilon)y_2} dy_2 = \frac{2}{(1 - p_\epsilon)}[\ln(y_2)|_{y_1}^{y_1/p_\epsilon}] = \frac{2 \ln(p_\epsilon)}{(p_\epsilon - 1)}. \tag{31}$$

For $p_\epsilon \le y_1 \le 1/2$ we have,

$$f_{y_1}(y_1) = \int_{y_1}^{1/2} \frac{2}{(1 - p_\epsilon)y_2} dy_2 = \frac{2}{(1 - p_\epsilon)}[\ln(y_2)|_{y_1}^{1/2}] = \frac{2 \ln(2y_1)}{(p_\epsilon - 1)}. \tag{32}$$

Thus, as a piecewise function the PDF is formally,

$$f_{y_1}(y_1) = \begin{cases} \frac{2 \ln(p_\epsilon)}{(p_\epsilon - 1)} & \text{for } 0 \le y_1 \le \frac{p_\epsilon}{2}, \\ \frac{2 \ln(2y_1)}{(p_\epsilon - 1)} & \text{for } \frac{p_\epsilon}{2} \le y_1 \le 1/2. \end{cases} \tag{33}$$

Now, the CDF is determined through integration,

$$F_{y_1}(y_1) = \int_0^{y_1} f_{y_1}(y_1) dy_1 = \begin{cases} \frac{2y_1 \ln(p_\epsilon)}{(p_\epsilon - 1)} & \text{for } 0 \le y_1 \le \frac{p_\epsilon}{2}, \\ \frac{2y_1(\ln(2y_1) - 1) + p_\epsilon}{(p_\epsilon - 1)} & \text{for } \frac{p_\epsilon}{2} \le y_1 \le 1/2. \end{cases} \tag{34}$$

We can integrate the CDF to get,

$$\int_0^{y_1} F_{y_1}(y_1) = \begin{cases} \frac{y_1^2 \ln(p_\epsilon)}{(p_\epsilon - 1)} & \text{for } 0 \le y_1 \le \frac{p_\epsilon}{2}, \\ \frac{4y_1^2(2\ln(2y_1)-3)+8y_1 p_\epsilon - p_\epsilon^2}{8(p_\epsilon - 1)} & \text{for } \frac{p_\epsilon}{2} \le y_1 \le 1/2. \end{cases} \tag{35}$$

Plugging all of this back into Equation 7 yields,

$$\hat{b}_i^* = \begin{cases} p_\epsilon + v_i^p \frac{2v_i^p \ln(p_\epsilon)}{p_\epsilon - 1} - \frac{(v_i^p)^2 \ln(p_\epsilon)}{p_\epsilon - 1}, \\ p_\epsilon + v_i^p \frac{2v_i^p (\ln(2v_i^p)-1)+p_\epsilon}{(p_\epsilon - 1)} - \frac{4(v_i^p)^2(2\ln(2v_i^p)-3)+8v_i^p p_\epsilon - p_\epsilon^2}{8(p_\epsilon - 1)}, \end{cases}$$

$$= \begin{cases} p_\epsilon + \frac{(v_i^p)^2 \ln(p_\epsilon)}{p_\epsilon - 1} & \text{if } 0 \le v_i^p \le \frac{p_\epsilon}{2}, \\ p_\epsilon + \frac{8(v_i^p)^2(\ln(2v_i^p)-1/2)+p_\epsilon^2}{8(p_\epsilon - 1)} & \text{if } \frac{p_\epsilon}{2} \le v_i^p \le \frac{1}{2}. \end{cases} \tag{36}$$

Since $b_i$ cannot be larger than 1, we cap the bidding function at one via,

$$b_i^* := \min\{\hat{b}_i^*, 1\}. \tag{37}$$

The utility gained by agent $i$ for using such a bidding function is,

$$u(b_i^*) = \begin{cases} v_i^d - b_i^* + \frac{2(v_i^p)^2 \ln(p_\epsilon)}{p_\epsilon - 1} & \text{for } 0 \le v_i^p \le \frac{p_\epsilon}{2}, \\ v_i^d - b_i^* + \frac{2(v_i^p)^2(\ln(2v_i^p)-1)+p_\epsilon}{(p_\epsilon - 1)} & \text{for } \frac{p_\epsilon}{2} \le v_i^p \le 1/2. \end{cases} \tag{38}$$

When this utility is larger than 0, the agent will participate otherwise the agent will not submit a model to the regulator. Finally, we can find the optimal compliance level by using Assumption 2,

$$s_i^* := M^{-1}(b_i^*). \tag{39}$$

$\square$

## D.4 PROOF OF COROLLARY 2

**Corollary 2** (Restated). *Under Assumption 2, let agents have total value $V_i$ and scaling factor $\lambda_i$ stem from Beta ($\alpha = \beta = 2$) and Uniform distributions respectively, with $v_i^d = (1-\lambda_i)V_i$ and $v_i^p = \lambda_i V_i$. Denote the CDF of the Beta distribution on $[0,1]$ as $F_\beta(x) = 3x^2 - 2x^3$. The optimal bid and utility for agents participating in* CIRCA *Equation 6 are,*

$$b_i^* := \min\{\hat{b}_i^*, 1\}, \quad \hat{b}_i^* = \begin{cases} p_\epsilon + \frac{3(v_i^p)^2(p_\epsilon^2 - 2p_\epsilon + 1)}{1 - F_\beta(p_\epsilon)} & \text{if } 0 \le v_i^p \le \frac{p_\epsilon}{2}, \\ p_\epsilon + \frac{8(v_i^p)^2\left(6(v_i^p)^2 - 8v_i^p + 3\right)+p_\epsilon^3(3p_\epsilon - 4)}{8(1 - F_\beta(p_\epsilon))} & \text{if } \frac{p_\epsilon}{2} \le v_i^p \le \frac{1}{2}, \end{cases}$$

$$u(b_i^*; \boldsymbol{b}_{-i}) = \begin{cases} v_i^d + \frac{6(v_i^p)^2(p_\epsilon^2 - 2p_\epsilon + 1)}{1 - F_\beta(p_\epsilon)} - b_i^* & \text{for } 0 \le v_i^p \le \frac{p_\epsilon}{2}, \\ v_i^d + \frac{v_i^p\left(8(v_i^p)^3 - 12(v_i^p)^2 + 6v_i^p + p_\epsilon^2(2p_\epsilon - 3)\right)}{1 - F_\beta(p_\epsilon)} - b_i^* & \text{for } \frac{p_\epsilon}{2} \le v_i^p \le 1/2. \end{cases}$$

*Participating agents submit models with compliance,*

$$s_i^* = \begin{cases} M^{-1}(b_i^*) > \epsilon & \text{if } u_i(b_i^*; \boldsymbol{b}_{-i}) > 0, \\ 0 \text{ (no model submission)} & \text{else.} \end{cases}$$

*Proof.* Similar to Corollary 1, we begin solving for $f_{v_i^p}(\cdot)$ using a change of variables. For the product of two random variables $v = x_1 \cdot x_2$, let $y_1 = x_1 \cdot x_2$ and $y_2 = x_2$. Inversely, $x_2 = y_2$ and $x_1 = y_1/y_2$. While $x_1$ and $x_2$ are independent, $x_1$ comes from a Beta distribution and $x_2$ from a Uniform one. The PDF and CDF of a Beta distribution, with $\alpha = \beta = 2$, on $[0,1]$ are defined as,

$$f_\beta(x) := 6x(1-x), \tag{40}$$

$$F_\beta(x) := 3x^2 - 2x^3. \tag{41}$$

Now, the PDF over $y_1, y_2$ is defined as,

$$f_{y_1, y_2}(x_1, x_2) = \left(\frac{6x_1(1-x_1)}{1 - F_\beta(p_\epsilon)}\right)\left(\frac{1}{1/2 - 0}\right) = \frac{12x_1(1-x_1)}{1 - F_\beta(p_\epsilon)}. \tag{42}$$

When using the change of variables this becomes,

$$f_{y_1,y_2}(y_1,y_2) = f_{y_1,y_2}(x_1,x_2)|J| = \frac{12y_1(1-\frac{y_1}{y_2})}{(1-F_\beta(p_\epsilon))y_2^2}, \quad |J| = \left| \begin{pmatrix} 1/y_2 & -y_1/y_2^2 \\ 0 & 1 \end{pmatrix} \right| = 1/y_2 \tag{43}$$

Marginalizing out $y_2$ (a non-negative value) yields,

$$f_{y_1}(y_1) = \frac{12y_1}{1-F_\beta(p_\epsilon)} \int_0^\infty \frac{1}{y_2^2} - \frac{y_1}{y_2^3} dy_2. \tag{44}$$

The bounds of integration depend upon the value of $y_1$. The change of variable to the $(y_1, y_2)$ space, where $0 \le y_1, y_2 \le 1/2$, results in a new region of possible variable values. This region is a triangle bounded by the three vertices: $(0,0)$, $(p_\epsilon/2, 1/2)$, and $(1/2, 1/2)$. Thus, the bounds of marginalization depend upon the value of $y_1$. For $0 \le y_1 \le p_\epsilon/2$ we have,

$$f_{y_1}(y_1) = \frac{12y_1}{1-F_\beta(p_\epsilon)} \int_{y_1}^{y_1/p_\epsilon} \frac{1}{y_2^2} - \frac{y_1}{y_2^3} dy_2 = \frac{12y_1}{1-F_\beta(p_\epsilon)} [-\frac{1}{y_2} + \frac{y_1}{2y_2^2}|_{y_1}^{y_1/p_\epsilon}]$$

$$= \frac{12y_1}{1-F_\beta(p_\epsilon)} [-\frac{p_\epsilon}{y_1} + \frac{p_\epsilon^2}{2y_1} + \frac{1}{y_1} - \frac{1}{2y_1}] = \frac{6(p_\epsilon^2 - 2p_\epsilon + 1)}{1-F_\beta(p_\epsilon)}. \tag{45}$$

For $p_\epsilon \le y_1 \le 1/2$ we have,

$$f_{y_1}(y_1) = \frac{12y_1}{1-F_\beta(p_\epsilon)} \int_{y_1}^{1/2} \frac{1}{y_2^2} - \frac{y_1}{y_2^3} dy_2 = \frac{12y_1}{1-F_\beta(p_\epsilon)} [-\frac{1}{y_2} + \frac{y_1}{2y_2^2}|_{y_1}^{1/2}]$$

$$= \frac{12y_1}{1-F_\beta(p_\epsilon)} [-2 + 2y_1 + \frac{1}{y_1} - \frac{1}{2y_1}] = \frac{6(4y_1^2 - 4y_1 + 1)}{1-F_\beta(p_\epsilon)}. \tag{46}$$

Thus, as a piecewise function the PDF is formally,

$$f_{y_1}(y_1) = \begin{cases} \frac{6(p_\epsilon^2 - 2p_\epsilon + 1)}{1-F_\beta(p_\epsilon)} & \text{for } 0 \le y_1 \le \frac{p_\epsilon}{2}, \\ \frac{6(4y_1^2 - 4y_1 + 1)}{1-F_\beta(p_\epsilon)} & \text{for } \frac{p_\epsilon}{2} \le y_1 \le 1/2. \end{cases} \tag{47}$$

Now, the CDF is determined through integration,

$$F_{y_1}(y_1) = \int_0^{y_1} f_{y_1}(y_1) dy_1 = \begin{cases} \frac{6y_1(p_\epsilon^2 - 2p_\epsilon + 1)}{1-F_\beta(p_\epsilon)} & \text{for } 0 \le y_1 \le \frac{p_\epsilon}{2}, \\ \frac{2y_1(4y_1^2 - 6y_1 + 3) + p_\epsilon^2(2p_\epsilon - 3)}{1-F_\beta(p_\epsilon)} & \text{for } \frac{p_\epsilon}{2} \le y_1 \le 1/2. \end{cases} \tag{48}$$

We can integrate the CDF to get,

$$\int_0^{y_1} F_{y_1}(y_1) = \begin{cases} \frac{3y_1^2(p_\epsilon^2 - 2p_\epsilon + 1)}{1-F_\beta(p_\epsilon)} & \text{for } 0 \le y_1 \le \frac{p_\epsilon}{2}, \\ \frac{8y_1\left(2y_1^3 - 4y_1^2 + 3y_1 + p_\epsilon^2(2p_\epsilon - 3)\right) + p_\epsilon^3(4 - 3p_\epsilon)}{8(1-F_\beta(p_\epsilon))} & \text{for } \frac{p_\epsilon}{2} \le y_1 \le 1/2. \end{cases} \tag{49}$$

Plugging all of this back into Equation 7 yields,

$$\hat{b}_i^* = \begin{cases} p_\epsilon + v_i^p \frac{6v_i^p(p_\epsilon^2 - 2p_\epsilon + 1)}{1-F_\beta(p_\epsilon)} - \frac{3(v_i^p)^2(p_\epsilon^2 - 2p_\epsilon + 1)}{1-F_\beta(p_\epsilon)}, \\ p_\epsilon + v_i^p \frac{2v_i^p(4(v_i^p)^2 - 6v_i^p + 3) + p_\epsilon^2(2p_\epsilon - 3)}{1-F_\beta(p_\epsilon)} - \frac{8v_i^p\left(2(v_i^p)^3 - 4(v_i^p)^2 + 3v_i^p + p_\epsilon^2(2p_\epsilon - 3)\right) + p_\epsilon^3(4 - 3p_\epsilon)}{8(1-F_\beta(p_\epsilon))}, \end{cases}$$

$$= \begin{cases} p_\epsilon + \frac{3(v_i^p)^2(p_\epsilon^2 - 2p_\epsilon + 1)}{1-F_\beta(p_\epsilon)} & \text{if } 0 \le v_i^p \le \frac{p_\epsilon}{2}, \\ p_\epsilon + \frac{8(v_i^p)^2\left(6(v_i^p)^2 - 8v_i^p + 3\right) + p_\epsilon^3(3p_\epsilon - 4)}{8(1-F_\beta(p_\epsilon))} & \text{if } \frac{p_\epsilon}{2} \le v_i^p \le \frac{1}{2}. \end{cases} \tag{50}$$

Since $b_i$ cannot be larger than 1, we cap the bidding function at one via,

$$b_i^* := \min\{\hat{b}_i^*, 1\}. \tag{51}$$

The utility gained by agent $i$ for using such a bidding function is,

$$u(b_i^*) = \begin{cases} v_i^d - b_i^* + \frac{6(v_i^p)^2(p_\epsilon^2 - 2p_\epsilon + 1)}{1-F_\beta(p_\epsilon)} & \text{for } 0 \le v_i^p \le \frac{p_\epsilon}{2}, \\ v_i^d - b_i^* + \frac{v_i^p\left(8(v_i^p)^3 - 12(v_i^p)^2 + 6v_i^p + p_\epsilon^2(2p_\epsilon - 3)\right)}{1-F_\beta(p_\epsilon)} & \text{for } \frac{p_\epsilon}{2} \le v_i^p \le 1/2. \end{cases} \tag{52}$$

When this utility is larger than 0, the agent will participate otherwise the agent will not submit a model to the regulator. Finally, we can find the optimal compliance level by using Assumption 2,

$$s_i^* := M^{-1}\left(b_i^*\right). \tag{53}$$

$\square$

## E  ADDITIONAL EXPERIMENTS

Within this section, we verify empirically that our computed PDF and CDFs in Corollaries 1 and 2 are correct. To accomplish this, we randomly sample and compute the product of $V_i$ and $\lambda_i$ fifty million times. We then plot the PDF and CDF of the resultant products and compare it with our theoretical PDF and CDF. The theoretical PDF and CDF for Corollary 1 are defined in Equations 33 and 34, while those for Corollary 2 are found in Equations 47 and 48. The results of these simulations, which validate our computed PDFs and CDFs, are shown in Figures 6 and 7. To ensure correctness, we perform testing on different values of $p_\epsilon$. As expected, our theory lines up exactly with our empirical simulations for both Corollaries as well as across varying $p_\epsilon$. We note that all experiments are computationally light, with all run locally on an M3 chip with 16GB of RAM.

Finally, we provide the full results of our case study in Section 6 in tabular form below.

Table 3: Equalized Odds as Minority Class Data Increases.

| Minority Class % | Mean Equalized Odds Score |
|---|---|
| 5% | 22.55 |
| 10% | 22.31 |
| 15% | 18.97 |
| 20% | 17.46 |
| 25% | 15.78 |
| 30% | 15.44 |
| 35% | 13.09 |
| 40% | 11.01 |
| 45% | 9.83 |
| 50% | 9.38 |

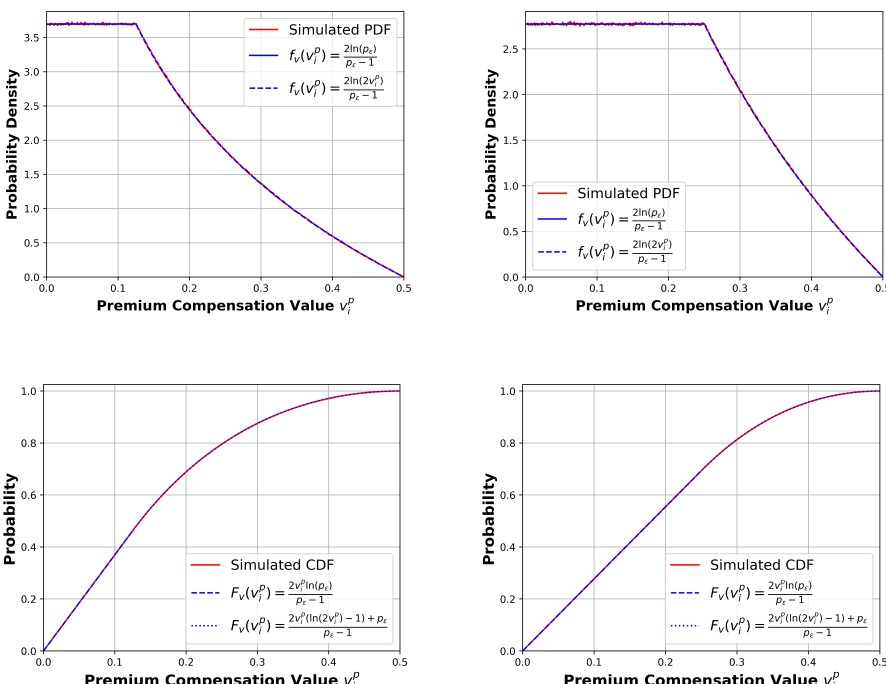

Figure 6: Numerical validation of our derivations for $f_v(v_i^p)$ and $F_v(v_i^p)$, where $v_i^p := V_i\lambda_i$, for $V_i$ and $\lambda_i$ coming from Uniform distributions (Corollary 1). The price of attaining $\epsilon$ is set as $p_\epsilon = 1/4$ (top row) and $p_\epsilon = 1/2$ (bottom row).

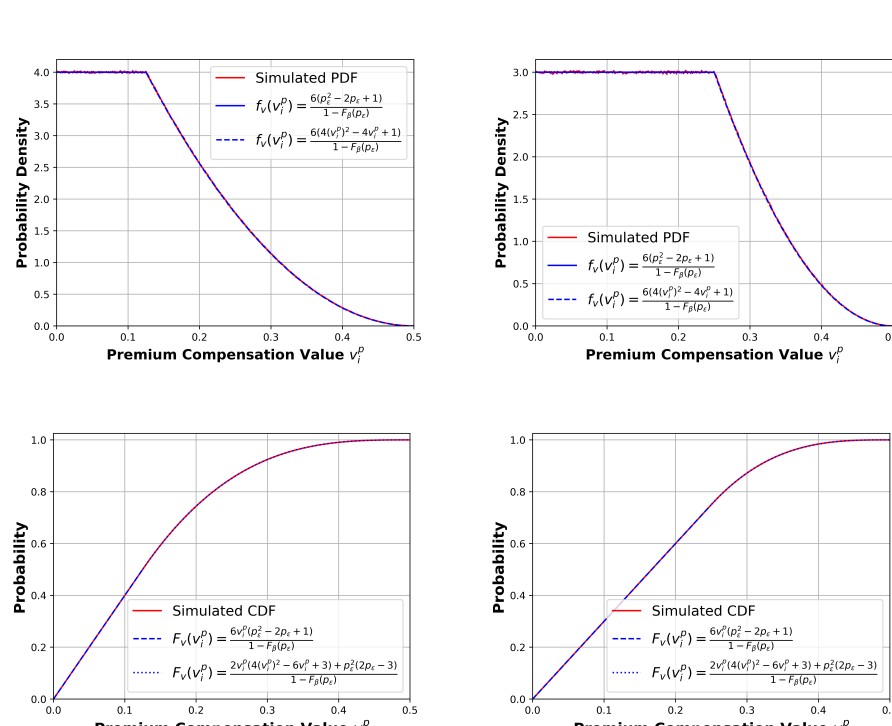

Figure 7: Numerical validation of our derivations for $f_v(v_i^P)$ and $F_v(v_i^P)$, where $v_i^p := V_i\lambda_i$, for $V_i$ coming from a Beta distribution and $\lambda_i$ from a Uniform distributions (Corollary 2). The price of attaining $\epsilon$ is set as $p_\epsilon = 1/4$ (top row) and $p_\epsilon = 1/2$ (bottom row).

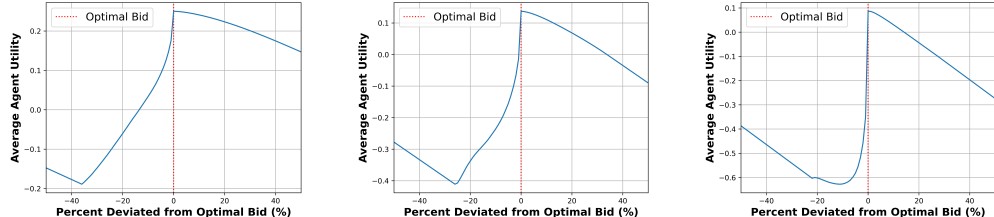

Figure 8: **Validation of Beta Nash Bidding Equilibrium.** Akin to the Uniform results, agent utility is maximized when agents follow the theoretically optimal bidding function shown in Equation equation 11. Across varying compliance prices, $p_\epsilon = 0.25$ (left), $0.5$ (middle), $0.75$ (right), agents attain less utility when they deviate from the optimal bid (red line) derived in Corollary 2.

# F   REPEATING CIRCA AUCTIONS

The current auction structure (Algorithm 1) expects agents to submit a single model trained solely for the upcoming auction. There is no expectation that the model will be reused for a future auction, or indication that the model has been submitted to a previous auction. Looking towards the future, we would like to design CIRCA to fit a repeatable auction structure, in which approved or rejected models may be resubmitted in subsequent auctions.

**Repeated Agent Utility**. Previously, in Algorithm 1, agents start the regulatory process with zero cost and value (*i.e.,* they are building their models from scratch). In repeating CIRCA auctions, agent cost and value are accumulated across all previous auction submissions. For example, if an agent trains its already-accepted model further to attain a higher compliance level $s_i$, its total accumulated training cost is $M(s_i)$. This agent's total value becomes the value its model gained from previous auction submissions plus any value gained from the current auction.

By allowing repeated CIRCA auctions, an agent is able to repeatedly submit its model for regulatory review. We note that repeated submissions decrease the value of model deployment; once an agent earns the reward for deploying their model, subsequent deployments of the same model with improved compliance levels can be realistically expected to earn less value than the initial deployment. We characterize this loss in value for repeated submissions with an indicator function in the utility function that only allows deployment value to be obtained once, on initial acceptance of a model. While we allow agents to win premium rewards across multiple auctions, we note that a regulator can curb this by either limiting the number of auction submissions per agent or the number of auctions held per year. We now define the repeated CIRCA auction utility of agent $i$, who has participated in $a - 1$ previous auctions, as:

$$u_{i,a}(b_i) = \left( \sum_{n=1}^{a} \nu_i^n \right) - b_i, \tag{54}$$

where $\nu_i^n$, the value gained at the $n^{th}$ auction model $i$ was submitted to, is formulated as:

$$\nu_i^n = \begin{cases} v_i^{d,n} \cdot 1_{(\text{if } \nu_i^{n-1} = 0)} & \text{if } b_j^n \geq p_\epsilon^n \text{ and } b_i^n < b_j^n \text{ randomly sampled bid } b_j^n, \\ v_i^{d,n} \cdot 1_{(\text{if } \nu_i^{n-1} = 0)} + v_i^{p,n} & \text{if } b_i^n \geq p_\epsilon^n \text{ and } b_i^n > b_j^n \text{ randomly sampled bid } b_j^n, \\ 0 & \text{if } n \leq 0. \end{cases} \tag{55}$$

The repeated CIRCA auction setup creates a unique property for models in training. If an agent intends to obtain a high compliance level, but an auction takes place mid-training, the agent is actually incentivized to submit their model early if they have a chance at winning the premium reward. Though the model may have a lower likelihood of earning the reward, there is no consequence for models failing to attain the premium reward. Gaining value is strictly beneficial to agents, and accumulated value helps offset the costs of training a model. This property only exists for the premium reward; the deployment reward can only be obtained once, thus there is no incentive to submit early to earn it.

**Repeated Optimal Bidding Function**. Using the same assumptions for single-auction CIRCA, namely Assumptions 1 and 2 along with private values, we can derive the bidding function for a rational agent under a repeated CIRCA auction setting. We follow an equivalent setup to Lemma 1 with regards to the valuation of rewards, giving us the cumulative distribution function for $v_i^p = V_i \lambda_i$ as $F_v(\cdot)$ and the probability distribution function as $f_v(\cdot)$.

From our definition of utility $u_{i,a}(b_i)$, we find that an agent $i$ that does not participate (*i.e.,* submitting $b_i = 0$) receives utility equal to $\nu_i^a$. However, since $b_i = 0$ will never be larger than $p_\epsilon$ (by definition), it must be true that $\nu_i^a = 0$ as well, since the model will never meet the required compliance threshold. Therefore, a non-participating agent will always receive non-negative utility.

$$u_{i,a}(0) = 0. \tag{56}$$

Following closely to the proof of Theorem 2 in Appendix D, we find that participating agents $i \in P$ (with $P$ defined in the previous proof) will now have a utility of,

$$u_{i,a}(b_i) = \nu_i^a + v_i^d \cdot 1_{(\nu_i^a = 0)} + v_i^p \mathbb{P}(b_i > b_j) - b(b_i), \quad b_j \sim \text{randomly sampled agent bid},$$

$$= \nu_i^a + v_i^d \cdot 1_{(\nu_i^a = 0)} + v_i^p F_v(b_i) - b(b_i). \tag{57}$$

Taking the derivative and setting it equal to zero yields,

$$\frac{d}{db_i} u_{i,a}(b_i) = v_i^p f_v(b_i) - b'(b_i) = 0. \tag{58}$$

As agents bid in proportion to their valuation, we solve the first-order conditions at $b_i = v_i^p$,

$$b'(v_i^p) = v_i^p f_v(v_i^p). \tag{59}$$

Note, at this point in the proof the bidding function calculation is now equivalent to the calculations found in Lemma 1. We can thus follow the same steps to reveal our optimal bidding function,

$$b(v_i^p) := p_\epsilon + v_i^p F_v(v_i^p) - \int_0^{v_i^p} F_v(z)dz, \tag{60}$$

which is equivalent to the optimal bidding function derived in Lemma 1.

As the optimal bidding function is equivalent, calculations for the Nash Bidding Equilibrium are also equivalent to those found in Corollary 1 and Corollary 2. The optimal bid and utility participating in CIRCA Equation 6 under the assumptions of Corollary 1 will thus be,

$$b_i^* := \min\{\hat{b}_i^*, 1\}, \quad \hat{b}_i^* = \begin{cases} p_\epsilon + \frac{(v_i^p)^2 \ln(p_\epsilon)}{p_\epsilon - 1} & \text{if } 0 \le v_i^p \le \frac{p_\epsilon}{2}, \\ p_\epsilon + \frac{8(v_i^p)^2(\ln(2v_i^p) - 1/2) + p_\epsilon^2}{8(p_\epsilon - 1)} & \text{if } \frac{p_\epsilon}{2} \le v_i^p \le \frac{1}{2}, \end{cases}$$

$$u_{i,a}(b_i^*; \boldsymbol{b}_{-i}) = \begin{cases} \nu_i^a + v_i^d \cdot 1_{(\nu_i^a = 0)} + \frac{2(v_i^p)^2 \ln(p_\epsilon)}{p_\epsilon - 1} - b_i^* & \text{if } 0 \le v_i^p \le \frac{p_\epsilon}{2}, \\ \nu_i^a + v_i^d \cdot 1_{(\nu_i^a = 0)} + \frac{2(v_i^p)^2(\ln(2p_\epsilon) - 1) + p_\epsilon}{p_\epsilon - 1} - b_i^* & \text{if } \frac{p_\epsilon}{2} \le v_i^p \le \frac{1}{2}. \end{cases}$$

Agents participating in CIRCA under Corollary 1 submit models with the following compliance,

$$s_i^* := \begin{cases} M^{-1}(b_i^*) > \epsilon & \text{if } u_i(b_i^*; \boldsymbol{b}_{-i}) > 0, \\ 0 \text{ (no model submission)} & \text{else.} \end{cases}$$

The optimal bid and utility participating in CIRCA Equation 6 under the assumptions of Corollary 2 will be,

$$b_i^* := \min\{\hat{b}_i^*, 1\}, \quad \hat{b}_i^* = \begin{cases} p_\epsilon + \frac{3(v_i^p)^2(p_\epsilon^2 - 2p_\epsilon + 1)}{1 - F_\beta(p_\epsilon)} & \text{if } 0 \le v_i^p \le \frac{p_\epsilon}{2}, \\ p_\epsilon + \frac{8(v_i^p)^2\left(6(v_i^p)^2 - 8v_i^p + 3\right) + p_\epsilon^3(3p_\epsilon - 4)}{8(1 - F_\beta(p_\epsilon))} & \text{if } \frac{p_\epsilon}{2} \le v_i^p \le \frac{1}{2}, \end{cases}$$

$$u_{i,a}(b_i^*; \boldsymbol{b}_{-i}) = \begin{cases} \nu_i^a + v_i^d \cdot 1_{(\nu_i^a = 0)} + \frac{6(v_i^p)^2(p_\epsilon^2 - 2p_\epsilon + 1)}{1 - F_\beta(p_\epsilon)} - b_i^* & \text{for } 0 \le v_i^p \le \frac{p_\epsilon}{2}, \\ \nu_i^a + v_i^d \cdot 1_{(\nu_i^a = 0)} + \frac{v_i^p\left(8(v_i^p)^3 - 12(v_i^p)^2 + 6v_i^p + p_\epsilon^2(2p_\epsilon - 3)\right)}{1 - F_\beta(p_\epsilon)} - b_i^* & \text{for } \frac{p_\epsilon}{2} \le v_i^p \le 1/2. \end{cases}$$

Agents participating in CIRCA under Corollay 2 submit models with the following compliance,

$$s_i^* = \begin{cases} M^{-1}(b_i^*) > \epsilon & \text{if } u_i(b_i^*; \boldsymbol{b}_{-i}) > 0, \\ 0 \text{ (no model submission)} & \text{else.} \end{cases}$$

# G  FUTURE WORK

While this work addresses key challenges in regulating AI compliance, several directions remain open for future exploration:

*(1) Model Evaluation:* Creating a realistic protocol for the regulator to evaluate submitted model compliance levels is important to ensure agents do not skirt around compliance requirements. While we leave this problem for future work, one possible solution is that agents can either provide the regulator API access to test its model or provide the model weights directly to the regulator. Truthfulness can be enforced via audits and the threat of legal action.

*(2) Extension to Heterogeneous Settings:* Extending our mechanism to heterogeneous scenarios, where evaluation data for agents and regulators differs, is a critical next step. Real-world data distributions often vary across contexts, and understanding how these variations affect both model

compliance and agent strategies will create a more robust regulatory mechanism. While explicit protocols or mathematical formulations are left as future work, we have a few ideas. One idea could be establishing a data-sharing framework between agents and the regulator, where each participating agent must contribute part of (or all of) its data to the regulator for evaluation. If data can be anonymized, then this would be a suitable solution. Another idea could be that the regulator collects data on its own, and can compare its distribution of data versus each participating agents' data distribution. If distributions greatly differ, then the regulator could collect more data or resort to the previous data-sharing method.

