# OpenReview forum: "Advancing Regulation in Artificial Intelligence: An Auction-Based Approach"
_ICLR.cc/2026/Conference — ICLR 2026 Conference Withdrawn Submission_

### Official Review · Reviewer_9KpU · 2025-10-24

**Soundness:** 1
**Presentation:** 2
**Contribution:** 1
**Rating:** 2
**Confidence:** 4

**Summary:**

This paper proposes a type of all-pay auction for AI regulation, whereby model-making agents submit models to a regulator for approval. In this scenario, the regulator enforces certain compliance thresholds while also rewarding those bidders who go above and beyond the threshold. The authors claim that this method provably incentivizes the deployment of compliant models and claim results showing that this method would boost compliance rates compared to baseline regulatory mechanisms.

**Strengths:**

- The paper goes to great lengths to support its arguments with economic theory and mathematical formulas.
- The paper is clearly written.

**Weaknesses:**

- Unfortunately, this seems like more of an economics or game theory paper than a paper that is appropriate for ICLR. The bulk of the potential contributions here could apply to virtually any other domain and their application to the domain of AI regulation feels like an afterthought.
- The use of agents, or their advantages here in this context, is never fully explained.
- There are a number of strong statements made about the AI regulation landscape in the Introduction that are not adequately supported. For example, this is a strong statement that has no citation: "This follows a consistent trend of well-deserved scrutiny towards the lack of AI regulation without providing an answer on how to develop rigorous and realistic mathematical frameworks to achieve AI regulation."
- There is little attempt to tie the work to the real-world landscape of AI regulation or to explain why, beyond purely economic principles, governments would be inclined to adopt this system. Similarly, little attempt is made to explain the practicalities of how the system would work in the modern AI landscape.
- There is no concrete comparison to other, alternative AI regulatory frameworks (including those being implemented in practice today).
- Technical details of the experiments were implemented are largely missing.
- In the experiment, the exclusive use of fairness as the single compliance metric feels inadequate, especially when it is only measured using a single face bias benchmark dataset. Thus far, most AI regulations have gone far beyond having a single requirement around fairness and bias as regards faces. Further, no AI regulation is cited or referred to when choosing fairness as the sole metric. A more interesting and realistic approach would have been to look at multiple compliance metrics simultaneously, citing existing AI regulations to make them feel relevant.

**Questions:**

- Were actual agents used in the experiments? If so, please describe them in greater detail.
- The use of agents is not fully explained; what advantage do they offer here?
- What are the practical obstacles to adoption of this system amid today's AI landscape?
- Amid the broad scope of contemporary AI regulation, why did you choose to focus on fairness, as measured with a single face bias benchmark, as your single compliance metric in the experiment?

---

### Official Review · Reviewer_juFf · 2025-10-30

**Soundness:** 3
**Presentation:** 3
**Contribution:** 2
**Rating:** 4
**Confidence:** 2

**Summary:**

This paper is inspired by game theory and it proposes a framework to regulate AI deployment through an auction-based compliance process. In this framework, the regulator defines both a compliance threshold—measured by performance on regulator-specified tasks—and the `cost'  (eg: inference cost) required to achieve it. Agents (such as AI developers) assess the total value they expect to receive from deploying their models and any potential regulatory rewards; they participate only if this value exceeds the threshold cost. Participating agents submit their models along with a bid that reflects their investment in achieving compliance. Models with bids below the threshold are excluded. The regulator then randomly pairs the remaining models and, in each pair, selects the model with higher compliance as the winner. Winning models receive both a deployment reward and a premium bonus, while losing models receive only the deployment reward.

**Strengths:**

- Interesting idea of applying game-theory to AI regulation compliance
- Easy to understand, simple formulation

**Weaknesses:**

See Question

**Questions:**

Does defining 'compliance' as a single scalar score (serving as a proxy for overall performance) limit the practical utility of this framework? In real AI regulation, models and developers may excel in different areas, and strengths can be distributed unevenly across tasks or compliance dimensions. With the current setup, where agents can observe and optimize for the compliance formula, there is a risk that developers focus their efforts on sub-tasks that are weighted more heavily, potentially neglecting others or “gaming” the aggregator. Alternatively, if all dimensions are treated uniformly, the framework may still disincentivize models that are exceptionally strong in some areas but weaker in others—or those that are broadly competent but not top-tier in any single metric. Could the authors clarify how their framework handles these cases, and whether it is adaptable to more multidimensional compliance objectives?

---

### Official Review · Reviewer_HESS · 2025-10-31

**Soundness:** 4
**Presentation:** 4
**Contribution:** 3
**Rating:** 6
**Confidence:** 3

**Summary:**

The paper proposes a new framework, CIRCA, for regulating AI in the form of an all-pay auction where enterprises submit models for approval and receive a commensurate reward if they submit the winning bid. The proposed framework results in the optimal strategy being to submit models that exceed the regulator’s compliance threshold. Validation of the Nash equilibrium for uniform and beta value distributions demonstrates that CIRCA results in agents bidding more compliant models that the baseline, Reserve Thresholding. Additionally, CIRCA encourages agents to be more participative than Reserve Thresholding.

**Strengths:**

CIRCA is predicated on a simple idea of adding a premium for a winning bid against a randomly selected agent to the agent utility and yields a clear improvement in compliance of the submitted models. The remainder of the work is dedicated to furnishing theoretical results demonstrating the benefits of the proposed framework in the form of the guarantee outlined in Theorem 2. The special cases of the uniform and beta value distributions serve to concretize the optimal agent strategy under the CIRCA framework. The experimental validation also does a good job of supporting the authors’ claims.

**Weaknesses:**

My primary concern with the paper is the feasibility of the premium offered to agents in a practical deployment of this framework. The benefit in terms of the additional margin in model compliance is linked directly to this premium. Given the sizeable costs in deployments of AI models, one would intuitively expect that the premium would have to be comparable to deployment costs, for CIRCA to offer a meaningful improvement over Reserve Thresholding. However, it is unclear if a regulator would be able to offer such premiums.

**Questions:**

**Questions:**

1.	When discussing value distributions (lines 204-210), the deployment and premium values are taken to be on the same scale; is this feasible in a practical implementation of this framework. If not, this should be acknowledged as a limitation in the discussion/conclusion.

**Suggestions:**

1.	Clarify what an all-pay auction is before delving into the details of CIRCA.

---

### Note · Authors · 2025-12-05

**Comment:**

We thank the reviewers for their service to the community.

We appreciate that the reviewers found our theoretical contributions worthy, albeit if they do not find it well-suited for ICLR. The goal of our paper is to get the ball rolling on a realistic regulatory framework for AI. We believe that the method we propose is mathematically-backed (the paper's main novelty) and able to be implemented in practice.  We will look for a venue that fits our auction-theory-based and game-theory inspired AI regulatory framework.

As a note, when detailing "agents" within our paper, this does not refer to AI agents or agentic models. Instead, this is a common term to denote "players" or "companies" that participate in our framework.

Best,
Submission12104 Authors

**Withdrawal Confirmation:**

I have read and agree with the venue's withdrawal policy on behalf of myself and my co-authors.